# Challenges and Opportunities in Developing Targeted Therapies for Triple Negative Breast Cancer

**DOI:** 10.3390/biom13081207

**Published:** 2023-08-01

**Authors:** Abygail G. Chapdelaine, Gongqin Sun

**Affiliations:** Department of Cell and Molecular Biology, University of Rhode Island, Kingston, RI 02881, USA; agchapdelaine@uri.edu

**Keywords:** triple negative breast cancer, targeted therapy, multi-driver oncogenesis, combination targeted therapy

## Abstract

Triple negative breast cancer (TNBC) is a heterogeneous group of breast cancers characterized by their lack of estrogen receptors, progesterone receptors, and the HER2 receptor. They are more aggressive than other breast cancer subtypes, with a higher mean tumor size, higher tumor grade, the worst five-year overall survival, and the highest rates of recurrence and metastasis. Developing targeted therapies for TNBC has been a major challenge due to its heterogeneity, and its treatment still largely relies on surgery, radiation therapy, and chemotherapy. In this review article, we review the efforts in developing targeted therapies for TNBC, discuss insights gained from these efforts, and highlight potential opportunities going forward. Accumulating evidence supports TNBCs as multi-driver cancers, in which multiple oncogenic drivers promote cell proliferation and survival. In such multi-driver cancers, targeted therapies would require drug combinations that simultaneously block multiple oncogenic drivers. A strategy designed to generate mechanism-based combination targeted therapies for TNBC is discussed.

## 1. Introduction

Breast cancer is the second leading cause of cancer-related deaths in American women [1,2]. About 15% of breast cancers [3,4,5] lack estrogen receptors (ER), progesterone receptors (PR), and HER2 [6,7]. Such cases are referred to as triple negative breast cancer (TNBC). TNBC shares ~75% overlap with basal-like breast cancer, and its cells tend to look like the epithelial cells of the outmost, basal layer of the breasts’ milk ducts. TNBC is more aggressive than other breast cancer subtypes [7,8,9], with a higher mean tumor size, higher tumor grade, the worst five-year overall survival, and the highest rates of recurrence and metastasis [10,11,12]. It is more common in premenopausal and African American women [13,14,15,16].

While ER^+^ and HER2^+^ breast cancer patients significantly benefit from targeted therapies blocking ER [17,18] and HER2 [12,19,20], a lack of hormonal targets in TNBC has left patients with minimal treatment options. As such, alternative targets need to be discovered and exploited for treating TNBC. Significant effort has been invested into finding such targets and developing targeted therapies for TNBC; however, they have yielded limited success, and no broadly effective targeted therapy has been approved for TNBC. Consequently, TNBC treatment still relies on chemotherapy, surgery, and radiation therapy [21,22]. While this approach has been successful in early-stage TNBCs, it is relatively ineffective in advanced-stage patients, reflected in the fact that metastatic TNBC has a 5-year survival rate of only 12% [23,24]. Finding targeted therapy treatments for TNBC patients is desperately needed, and the struggle in developing such an approach epitomizes the limited reach of targeted cancer therapies in general. In 2018, only 8.33% of all US cancer patients were genomically eligible for targeted therapies and only 4.9% benefited from the treatment [25]. 

In this article, we review the current landscape of targeted therapy for TNBC, with a focus on the challenges in and prospects for developing targeting kinase-based signaling. We will review the evidence that supports TNBCs as multi-driver cancers and discuss approaches for identifying targeted drug combinations that simultaneously block multiple oncogenic drivers.

## 2. Approved Targeted Therapy Options for TNBC Patients 

Currently, treatment for TNBC primarily relies on the combination of surgery, chemotherapy, and radiation therapy, supplemented with inhibitors against poly-ADP ribose polymerases (PARP), immunotherapy with checkpoint inhibitors, and an antibody-drug conjugate inhibitor for topoisomerase [26]. PARP inhibition, immunotherapy using checkpoint inhibitors, and the topoisomerase inhibitor target specific molecular processes in cancer cells, making them targeted therapies (Table 1). These targeted therapies have their respective genetic requirements and can benefit sub-populations of TNBC patients. 

### 2.1. PARP Inhibition

PARPs are an enzyme family important for base excision repair and single-stranded DNA break repair [27,28]. Blocking PARPs causes the accumulation of single-stranded DNA breaks, ultimately leading to double stranded DNA breaks. Cells under PARP inhibition become more reliant on homologous recombination for fixing the single and double stranded breaks [27]. PARP inhibitors work best in patients with *BCRA1* or *BRCA2* mutations due to their role in homologous recombination. If a cancer is also deficient in BRCAs, the cancer cells will not be able to carry out homologous recombination, leading to cell death [29]. The PARP inhibitors olaparib [30] and talazoparib [31] are effective in treating TNBC patients who have mutations in *BRCA1* or *BRCA2*. They have also been approved for treating HER2-negative breast cancer patients who meet the same genetic criteria [32,33]. While PARP inhibition is successful in patients who meet this criteria, only about 15% of TNBC patients have mutations in *BRCA1* or *BRCA2* [34], so a majority of TNBC patients are not eligible for PARP inhibition treatment.

### 2.2. Immunotherapy with Checkpoint Blockade

Another process successfully exploited for TNBC targeted therapy is the interaction between cancer cells and the T-cells of the immune system. The binding between programmed cell death protein 1 (PD-1) from T-cells and the PD-1 ligand (PD-L1) on the surface of cancer cells, often referred to as immune checkpoint, prevents the destruction of the cancer cells by the immune system. Blocking the interaction between PD-L1 and PD-1 enhances immune-killing of cancer cells [35]. For the immune system to effectively attack the cancer cells, the cancer cells need to have a high tumor mutational burden due to microsatellite instability or defects in mismatch repair, rendering the cells sufficiently “foreign” to be attacked by the immune system [36,37,38]. For advanced TNBC patients in which the cancer cells express the PD-L1 protein, pembrolizumab (brand name: Keytruda) [39], a PD-1 monoclonal antibody, and atezolizumab (brand name: Tecentriq) [40], a PD-L1 monoclonal antibody, have, in combination with chemotherapy, been approved; however, the frequency of genetic defects resulting in microsatellite instability and defects in mismatch repair is very low in TNBC [41,42], limiting the reach of immunotherapy for TNBC patients.

### 2.3. Antibody-Drug Conjugate Sacituzumab Govitecan

Sacituzumab govitecan (also referred to as sacituzumab govitecan-hziy or Trodelvy) is an antibody-drug conjugate in which a humanized monoclonal antibody against trophoblast cell surface antigen 2 (Trop-2) is coupled to SN-38, an inhibitor for topoisomerase I [43,44] with a hydrolysable linker. Trop-2 is a transmembrane calcium signal transducer over-expressed in many epithelial cells [45], including TNBC cells [46]. Sacituzumab govitecan binds to Trop-2 on the TNBC cell surface and helps deliver SN-38 to the tumor cells. The antibody-SN-38 conjugate can be hydrolyzed to release SN-38 into the tumor microenvironment or be internalized into the cancer cells and release SN-38 inside tumor cells. SN-38 inhibits topoisomerase activity and kills the cancer cells. Large scale clinical trials demonstrated its effectiveness against TNBC. In a phase I/II trial (NCT01631552) including 108 TNBC patients that had a median of 3 previous therapies, 3 had complete responses and 33 had partial responses, resulting in a response rate of 33.3% [47]. A phase III clinical trial (NCT02574455) conducted on patients with relapsed or refractory metastatic TNBC compared treatments between sacituzumab govitecan and single-agent chemotherapy of the physician’s choice. The trial revealed that sacituzumab govitecan treatment had better outcomes than chemotherapy in median progression-free survival (5.6 versus 1.7 months), median overall survival (12.1 versus 6.7 months), and objective response rate (35% versus 5%) [48]. The treatment is approved by the FDA for patients with unresectable locally advanced or metastatic TNBC who have received two or more prior systemic therapies, at least one of them for metastatic disease [49].

## 3. Numerous Potential Drug Targets in Cancer Cell Signaling Have Emerged in TNBC 

While targeted therapies blocking PARPs, immune checkpoints, and topoisomerase I are effective for TNBC patients meeting their respective requirements, the majority of TNBC patients do not meet these requirements. Traditional targeted therapies blocking oncogenic signaling are still urgently needed for most TNBC patients. Among the eight cancer hallmarks [50,51], at least six are supported by oncogenic signaling: sustaining proliferative signaling, evading growth suppressors, resisting cell death, inducing angiogenesis, activating invasion and metastasis, and reprograming energy metabolism. Thus, blocking cancer cell signaling is a central strategy for cancer targeted therapy. Targeted therapy blocking signaling pathways has been a successful approach for numerous cancers, such as targeting BCR-Abl in chronic myeloid leukemia (CML) [52,53], HER2 in HER2-positive breast cancer [19,54], epidermal growth factor receptor (EGFR) in non-small cell lung cancer (NSCLC) [55,56], Kit in gastrointestinal stromal tumors (GIST) [57,58], and BRAF in BRAF V600E-containing melanoma [59,60]. These successful examples demonstrate that blocking signaling drivers is a highly effective approach for targeted cancer therapy.

Analysis of TNBC tumor samples and cells have identified numerous oncogenic mutations, giving rise to the hope that blocking these signaling targets would yield clinical benefits for TNBC treatment. The function of tumor suppressor TP53 is lost in most TNBC tumors due to mutation (84%) or other pathway inactivating events, such as gain of MDM2 (14%) [3]. Up to 15–25% of TNBCs have mutations in *BRCA1* or *BRCA2* [34,61,62,63]. The phosphatidylinositol 3-kinase (PI3K) pathway is frequently activated in TNBC due to *PIK3CA* mutation (9%) and amplification (49%), *PTEN* mutation/loss (35%), and *INPP4B* loss (30%) [3]. TNBC is also associated with high genomic instability and aneuploidy, leading to the amplification and deletion of signaling proteins. Frequently amplified genes include receptor protein tyrosine kinases (rPTK), including *EGFR*, fibroblast growth factor receptors (*FGFR*s), insulin-like growth factor receptor 1 (*IGF-1R*), *Kit*, *Met*, and *PDGFRA*, and signaling proteins in the mitogen-activated protein kinase (MAPK) pathway, including *KRAS* (32%) and *BRAF* (30%) [3,64]. Src kinase is also frequently upregulated in TNBC [65,66,67,68,69,70,71].

The association between the activation of the PI3K pathway and cancer development is well established and extensively reviewed [72,73]. Class I PI3Ks phosphorylate the 3-OH group of phosphatidylinositol 4,5-bisposphate (PIP_2_) to generate phosphatidylinositol 3,4,5-bisposphate (PIP_3_), which binds to Akt to promote its phosphorylation and activation. PTEN is a lipid phosphatase that hydrolyzes PIP_3_ to PIP_2_ to down-regulate Akt activation. Akt then phosphorylates a series of proteins to regulate metabolism, stimulate cell proliferation and survival, and prevent cell death. Mutations of PI3K, especially p110α encoded by *PIK3CA*, and mutation/loss of *PTEN*, are frequently observed in cancer broadly [74] and have been shown to activate PI3K/Akt to promote cell proliferation and prevent cell death [75], causing cancer cells to become resistant to treatment by chemotherapy [76]. Thus, the constitutive activation of the PI3K pathway makes it a key cancer-causing pathway and blocking its function has long been recognized as an anti-cancer target in general, as well as in TNBC [77].

Although signaling proteins in the mitogen-activated protein kinase (MAPK) pathway are not frequently mutated, some TNBC cancers do contain oncogenic mutations in *KRAS* [78] or *BRAF* [79,80,81]. *KRAS* and *BRAF* genes are also frequently amplified in TNBC. The MAPK pathway is also activated by receptor tyrosine kinases, such as EGFR, which is often overexpressed in TNBC [82,83]. Furthermore, the MAPK pathway is a key growth promoting pathway in cancer broadly. Thus, there is a strong rationale for targeting the MAPK pathway in TNBC [84]. 

Src protein tyrosine kinase is one of the most ubiquitous oncogenic drivers [85] and is strongly associated with TNBC development. A large portion of TNBC tumors overexpress Src and contain activated Src kinase [68,86,87]. Numerous studies have demonstrated that blocking Src activity inhibits the proliferation of many TNBC cell lines [68,70,88,89,90,91,92,93,94,95,96]. Src kinase activity has been shown to promote cell proliferation, survival, and metastasis [85,97,98].

## 4. Clinical Trials and Case Studies of Signaling-Based Targeted Therapies in TNBC 

Numerous clinical trials have targeted the PI3K pathway, the MAPK pathway, EGFR, and Src (Figure 1). Most of the clinical trials did not report results due to disappointing outcomes. The results that are available for examination offer insight into the roles that these pathways play in TNBC development and the potential of targeting these pathways in future efforts. Most of the results are collected from the website https://clinicaltrials.gov/ (accessed on 10 May 2023) and journal publications.

### 4.1. Clinical Trials Targeting the PI3K Pathway in TNBC 

In the clinicaltrials.gov database of the National Library of Medicine, a search for the PI3K pathway (keywords: PI3K/AKT and triple negative breast cancer), identified 46 clinical trials. Most of these trials (37 of 46) combined a PI3K pathway inhibitor with some other form of therapy, mainly chemotherapy. Many of the completed studies (32 studies) did not report results due to futility, early termination, or lack of efficacy. Of those that published their results, most saw minimal improvements in patients treated with a PI3K pathway inhibitor as a monotherapy. For example, a phase II clinical trial studying the effectiveness of the PI3K inhibitor BKM120 (NCT01629615) on metastatic TNBC patients reported that no patient achieved a complete response (CR: disappearance of tumor) nor partial response (PR: tumor shrinkage of 30%). A significant portion of patients (17/50 = 34%) observed stable disease (SD: between 30% shrinkage and 20% growth), while 12% remained stable for over 4 months (Table 2). Similar results were obtained with alpelisib, another PI3K inhibitor (NCT02506556) [99]. This study included 10 TNBC patients with PI3K pathway mutations that had previously been heavily treated for their cancer. None of the patients achieved PR nor CR; however, 5 of 10 treated patients achieved stable disease [99]. Due to the lack of complete or partial responses, the study stopped recruiting TNBC patients. Interestingly, the same study reported 10 of 26 patients in the ER+ cohort achieved partial responses due to the same treatment [99].

Another trial targeting the PI3K pathway (NCT01277757) [100] tested the clinical benefits of the Akt inhibitor MK-2206 on metastatic breast cancer patients who had tumors with PIK3CA/Akt mutations and/or PTEN loss/mutation. The study cohort included 9 TNBC patients. The trial observed expected but manageable toxicity, such as fatigue and rash, but demonstrated limited clinical and pharmaco-dynamic activity as a monotherapy. Among the nine TNBC patients, one patient achieved a 6-month progression-free survival. The study was stopped early due to futility. 

These clinical trials suggest that targeting the PI3K pathway alone is unlikely to be successful in treating breast cancer [101]. The lack of PRs and CRs across various trials suggests that the PI3K pathway is not the sole oncogenic driver in most TNBC tumors. This is consistent with in vitro results in which no TNBC cell line model has been shown to be solely driven by PI3K pathway mutations. While many of these clinical trials were deemed unsuccessful after failing to result in PR or CR, it is important to note that a significant portion of patients observed stable disease (SD) for a significant period of time, indicating that blocking the PI3K pathway inhibited TNBC tumor growth. Thus, a combination with drugs targeting other pathways is likely necessary for effectively treating TNBC. 

### 4.2. Clinical Trials Targeting EGFR in TNBC

EGFR is overexpressed in more than 50% of TNBC patients, and it is one of the most important regulatory components for cell growth, proliferation, survival, and differentiation [102]. Its prevalence in TNBC patients and its oncogenic properties make it a promising candidate for targeted therapy, yet it has not been shown to have sufficient benefit in clinical trials as a monotherapy. 

Several phase II clinical trials using an EGFR inhibitor (gefitinib or erlotinib) as monotherapy on advanced breast cancers (not necessarily TNBC) found minimal benefits from treatment, reporting PR rates of 0–3% and no reported CRs [103,104,105]. While these results are disappointing, one of the studies reported that 12 of 31 advanced breast cancer patients (38.7%) were assessed as having SD, 3 of which were stabilized for at least 6 months [104]. 

### 4.3. Clinical Trials Targeting Src in TNBC

Preclinical studies have established the broad and important role of Src in TNBC development [98]. Three clinical trials using dasatinib (a Src inhibitor) as a monotherapy for TNBC have been completed with reported results. In a phase II trial (NCT02720185), five nuclear EGFR-positive TNBC patients were treated with dasatinib at 100 mg once daily as a neoadjuvant therapy for 7–10 days before planned surgery. One of the 5 patients had a pathological complete response with no evidence of disease at a 24 month follow-up; however, the study was terminated early due to COVID/low enrollment [106]. In another neoadjuvant phase II trial (NCT00817531), out of 22 patients, two patients (9%) had a partial response and 15 patients (68%) had stable disease. Adversely, five patients (22%) had disease progression [107]. In the final phase II trial (NCT00371254) [108], 44 unselected patients with locally advanced or metastatic TNBC were treated with dasatinib at 100 mg twice daily (BID) (23 of 44) or 70 mg BID (21 of 44). Of the 43 response-evaluable patients, 7 patients discontinued due to toxicity, 2 patients had partial responses lasting 14 and 58 weeks, 12 had stable disease (2 of which continued for more than 16 weeks), and 22 had disease progression. The median progression-free survival was 8.3 weeks. 

Similar clinical response patterns were observed when targeting Src or the PI3K pathway. Both treatments did not achieve the desired rates of CR and PR, but many patients achieved SD. These results are consistent with preclinical studies that support Src and the PI3K pathway as major growth promoters in TNBC. The lack of efficacy in achieving CR and PR by dasatinib and PI3K pathway inhibitors is also consistent with preclinical results. To achieve CR or PR, a treatment must kill cells in a pre-existing tumor. In preclinical studies, inhibiting the PI3K pathway or Src alone inhibits cell proliferation but does not decrease the number of TNBC cells. These results suggest that TNBC tumors are likely driven by multiple oncogenic drivers, and inhibition of any one driver may not be sufficient for TNBC treatment. 

### 4.4. Clinical Trials and Case Studies Targeting the MAPK Pathway in TNBC

There have been less than 20 clinical trials targeting the MAPK pathway for TNBC treatment, and there were no instances of monotherapy treatments that reported results. BRAF (a component of the MAPK pathway) mutations are rare in breast cancer and TNBC, and no clinical trials have directly targeted BRAF for TNBC treatment. With that being said, several BRAF V600E-driven TNBC cases and BRAF-targeting treatments have been reported [79,80,81], offering insightful glimpses into the challenges and opportunities in targeted therapy for TNBC.

Pircher et al. reported [81] that a 38-year-old female TNBC patient developed multiple lung metastases two years after neoadjuvant chemotherapy, mastectomy, and adjuvant chemotherapy. The patient went through multiple rounds of additional chemotherapy, irreversible electroporation to treat the lung metastasis, and surgical resection of a breast recurrence during the subsequent 12 months. The lung metastasis still progressed, and chemotherapy had to be stopped due to strong side effects. Out of options for additional treatments, next-generation sequencing on a lung biopsy revealed that the tumor contained a BRAF V600E mutation. V600E-containing BRAF has been an effective target in melanoma treatment [59,60], and the patient was subsequently treated with vemurafenib (a BRAF inhibitor) at 720 mg orally twice daily. The patient showed partial remission within three months with limited side effects. At the time of the report (19 months after the therapy started), the lung metastases remained radiologically stable, and the patient remained in good clinical condition. 

Another case of a TNBC patient with a BRAF V600E mutation was reported by Wang et al. [79]. The 60-year-old patient was first treated with right breast mastectomy and axillary lymph node dissection, followed by adjuvant chemotherapy and radiotherapy. Unfortunately, the patient still developed multiple new pulmonary nodules and lymph node metastases two months later. Additional chemotherapy resulted in a 7-month progression-free survival, followed by progressive disease. Next-generation sequencing revealed mutations in BRAF (V600E), PI3K, P53, and other genes in the primary tumor, so the patient received treatment with vemurafenib and paclitaxel. Due to toxicity, this was reduced to vemurafenib monotherapy. While some pulmonary and lymph node lesions showed regression, others showed concomitant progression. Sequencing revealed that the progressive lesions had acquired additional mutations in PDGFRB, NF2, GRM3, MLH1, FOXA1, LRP1B, and AR amplification compared to pretreatment. The patient eventually died of multiple organ failures 12 months after the initial advanced diagnosis. It appears that additional oncogenic mutations made these progressive lesions resistant to the BRAF-targeted treatment.

The third TNBC patient with a BRAF V600E mutation was a 57-year-old woman with metastatic TNBC and chemotherapy-refractory massive pleural effusion [80]. After failures of radiation therapy and chemotherapy to prevent disease progression, next-generation sequencing identified multiple oncogenic mutations, including BRAF V600E, PIK3CA H1047R, CDKN2A R58X, and TP53 W136X. The patient was treated with a combination of the BRAF inhibitor dabrafenib (150 mg twice daily) and the Mek inhibitor trametinib (2 mg once daily). The patient exhibited decreases in swelling and pain, a decrease in pleural fusion, a reduction in the size of the axillary lymph nodes, and a general improvement in conditions for six weeks. Despite these improvements, the patient eventually developed another subcutaneous tumor and died 12 weeks after initiating the dabrafenib/trametinib treatment. 

All three case reports support BRAF as a valid treatment target in TNBC harboring a V600E mutation. These case studies also make it clear that additional oncogenic driver mutations would confer intrinsic or acquired resistance to BRAF-based treatments. Identifying the additional drivers and developing drug combinations blocking both BRAF and other activated pathways are likely necessary to overcome such intrinsic or acquired resistance to treat TNBC patients effectively. 

The results of some clinical trials targeting the PI3K pathway and Src discussed above have been reported on the website https://clinicaltrials.gov/ (accessed on 10 May 2023). Those results are summarized in Table 2.

### 4.5. Clinical Trials Utilizing Combination Treatments in TNBC

While signaling-based monotherapy has been largely disappointing in a clinical setting, multiple clinical trials implementing combination-therapy have had improved outcomes for TNBC patients. Two general approaches of combination have been tried: combining a targeted therapy agent with chemotherapy or combining two targeted therapies.

A phase II clinical trial examining the benefits of adding ipatasertib (an AKT inhibitor) to paclitaxel (a chemotherapy agent) found that patients in the ipatasertib + paclitaxel group had a median progression free survival (PFS) of 6.2 months compared to 4.9 months in the placebo + paclitaxel group [109]. Similar benefits from adding an AKT inhibitor to chemotherapy were also observed in similar clinical trials [110,111]. Another phase II clinical trial (NCT00463788) of cetuximab (an anti-EGFR monoclonal antibody) in combination with cisplatin (a chemotherapy agent) found that patients in the combination group had an overall response rate (CR + PR) of 20% compared to 10% in the group that only received chemotherapy [112]. The combination group also had a longer median PFS than the chemotherapy group (3.7 vs. 1.5 months). Two additional clinical trials (NCT00633464 and NCT01097642) examined the benefits of adding cetuximab to a different chemotherapy agent, ixabepilone, in treating TNBC patients, but neither observed consistent benefits for the combination over ixabepilone alone. 

Several clinical trials combined MAPK pathway inhibitors with PI3K pathway inhibitors. Most did not report results; however, one study that stands out is a phase II clinical trial that observed the benefits of combining GSK2141795 (an AKT inhibitor targeting the PI3K pathway) and trametinib (a MEK inhibitor targeting the MAPK pathway) (NCT01964924) [113]. Patients began the trial with trametinib as a monotherapy. If no clinical benefit was observed, GSK2141795 was added to their regimen. Of the 37 patients that started monotherapy with trametinib, 51% (19 patients) had disease progression and started the combination therapy. The clinical benefit rate (CBR: CR + PR + SD) of the trametinib monotherapy was 21.6%, while the CBR of the combination therapy was 31.6%. The median PFS for trametinib was 7.7 weeks (4.43 to 8.29), and the PFS for the combination was 7.86 weeks (5.86 to 13.86). The combination was well tolerated with no patients requiring dose modifications and/or dose delays. These results suggest that blocking multiple signaling pathways may be beneficial for TNBC treatment, further supporting the idea that multiple signaling pathways may be contributing to TNBC development simultaneously. 

As accumulating evidence supports the multi-driver nature of TNBC, combination targeted therapy may be necessary for effectively treating TNBC. This approach is still in its infancy, as the combination is empirically selected rather than chosen based on the molecular mechanisms of a given cancer. In general, the benefits of current combination targeted cancer therapies are mostly derived from different subpopulation of patients benefiting from different components of a combination, rather than patients benefiting from the synergy and additivity of the drug combination [114]. Synergistic combination targeted therapy will be dependent on identifying the driving mechanisms of a given cancer and formulating combinations that simultaneously block multiple drivers. TNBCs appear to be multi-driver cancers that would benefit from this approach.

### 4.6. What Lessons Can Be Learned from Clinical Trials of TNBC Targeted Therapies? 

While the results from clinical trials have been disappointing, they do offer several important insights that may help future efforts in developing targeted therapies for TNBC. Below are several observations that become evident from this analysis.

First, targeting individual signaling pathways is unlikely to be sufficient to achieve significant clinical benefits due to the multi-driver nature of TNBC. Of all the TNBC cell line models, only one, DU-4475, has been shown to be a mono-driver cancer cell line [115]. Correspondingly, TNBC tumors with a BRAF V600E mutation, like DU-4475, also respond favorably to BRAF/Mek targeted treatment. No other TNBC cell line can be killed by blocking a single driver [115]. Thus, they are likely dependent on multiple drivers. Such multi-driver cancers require drug combinations simultaneously blocking all drivers to achieve significant therapeutic benefits. 

Second, preclinical results and clinical results are generally consistent. There has been an apparent discrepancy between preclinical promise and clinical ineffectiveness of targeted therapy for TNBC. However, a review of the clinical and preclinical studies indicates that the clinical and preclinical responses are consistent with each other. For example, the Src inhibitor dasatinib inhibits TNBC cell growth in vitro, inhibits TNBC tumor growth in animal models, and inhibits TNBC tumor growth in TNBC patients, resulting in stable disease in a significant portion of patients. Even the fact that dasatinib treatment does not result in CR and PR in patients is also predicted in preclinical studies, as few studies have demonstrated that dasatinib kills TNBC cells in vitro or eliminates/shrinks a TNBC tumor in animal models. 

Third, a biomarker–drug response relationship needs to be established in preclinical studies to enable biomarker-guided patient selection in the clinical setting. TNBC clinical trials rarely select patients guided by biomarkers, because the relationship between biomarkers and drug responses is not well-established in preclinical studies. This is an especially serious problem for a notoriously heterogeneous cancer like TNBC. Because of this heterogeneity, any given oncogenic driver is only operable in a small portion of TNBC patients. Biomarker-guided matching between patients and treatment will greatly improve the response rate. 

Finally, effective strategies of combination targeted therapy are needed to block multi-driver oncogenesis in TNBC. With most TNBC likely being dependent on multiple drivers for proliferation and survival, it is necessary to use drug combinations to block multiple drivers for effective targeted therapy. Although some clinical trials have used drug combinations, these combinations are not selected based on the oncogenic mechanisms. Such treatments lack precision and would be compromised by non-responsive patients. Strategies for mechanism-based combination targeted therapy are needed.

## 5. Developing Combination Targeted Therapy for TNBC

As accumulating evidence supports the concept of multi-driver oncogenesis in TNBC, it becomes evident that drug combinations simultaneously blocking multiple drivers would be necessary to treat TNBC. In this section, we review the recent efforts at identifying mechanism-based drug combinations for TNBC models in vitro. 

### 5.1. Multi-Driver Oncogenesis and Combination Targeted Therapy 

As discussed above, there is strong evidence that most TNBC cancers contain multiple oncogenic drivers. The likely multi-driver nature of TNBC is not surprising, as most cancers are multi-driver cancers [116,117,118,119,120,121]. Cancer development is an evolutionary process of selecting cells with growth and survival advantages in a tumor micro-environment [116,122,123]. Multiple growth and survival drivers would confer such a selective advantage. A recent study [121] of 7664 tumors of 29 types revealed that a tumor carries four driver mutations on average, but the number varies widely (from 1 to >10) among cancer types. Frequent mutations and amplification in rPTKs, PI3K and MAPK pathways, and Src upregulation in TNBC likely activate multiple drivers [3]. The lack of success of monotherapy against a broad array of oncogenic drivers also supports the multi-driver nature of TNBC. 

Identifying drug combinations for cancers mostly relies on empirical screening [124,125,126,127]. For example, Wali et al. [125] assessed 768 drug combinations between 128 drug candidates and six FDA-approved drugs on TNBC cells. Such studies can identify effective drug combinations; however, an insufficient number of combinations, an incomplete coverage of prospective drivers, and the absence of mechanistic considerations limit the potential of this approach. Some clinical trials also empirically formulate drug combinations [128] but lack a strategy for mechanism-based formulations. A 2017 analysis [114] of current drug combinations in clinical trials and patient-derived xenograft animal models revealed that most of the benefits of combination cancer therapies were due to different patient subgroups benefiting from different components of a combination rather than from synergy or additivity of the combination on individual patients. Truly harnessing the power of combination targeted therapy is dependent on identifying the oncogenic drivers and developing mechanism-based drug combinations to block all drivers in TNBC.

### 5.2. Current Pharmacological Models Are Not Suitable for Analyzing the Drug Response of Multi-Driver Cancers 

Developing drug combinations that simultaneously block multiple drivers is a unique challenge for cancer. Modern drug discovery is largely focused on finding and optimizing drugs against individual targets. The current pharmacological analysis is based on a one-drug-one-target paradigm coded in several versions of the Hill equation [129]. It is often expressed as follows:I = I_max_ × D^n^/((IC_50_*)^n^ + D^n^)) 

In this equation, the inhibitory effect (I) is a function of maximal inhibition (I_max_), drug concentration (D), half inhibitory concentration (IC_50_), and inhibitory slope or the Hill Co-efficient (n). The IC_50_ reflects the affinity between the drug and the target, and “n” measures the cooperativity in binding. When n is above 1, there is positive cooperativity, famously exemplified by O_2_ binding to hemoglobin. When n is below 1, it is assumed to be “negative cooperativity”, which remains a mechanistic enigma in modern pharmacology [130]. Because Hill equation-based pharmacology interprets drug response based on a one-drug-one-target paradigm, it is not adequate in characterizing the effects of a kinase-based targeted drug on multi-driver cancer cells, where one drug can exert its effects by inhibiting multiple targets [78,131].

Accumulating evidence indicates that cancer cell drug responses cannot be readily described by the pharmacological models represented by the Hill equation. Many cancer cells display unusually shallow response curves when treated by targeted drugs, especially those blocking the Akt/PI3K/mTOR pathway [130]. The shallow inhibition curves fall into the “negative cooperativity” category that has no ready mechanistic explanation [130]. Another report found that 28% of cancer drug responses are multiphasic [132]. A new paradigm for analyzing such complex responses was recently developed.

### 5.3. A Strategy for Mechanism-Based Targeted Drug Combination

Based on the analysis of shallow and multiphasic inhibition, a new mathematical model for analyzing the effects of targeted drugs on multi-driver cancers was recently developed. Shen et al. determined that shallow inhibition is biphasic in nature consisting of a potent target-specific inhibition and less potent off-target inhibition. This is a unique feature for the effects of targeted drugs on multi-driver cancer cells, as mono-driver cancer cells are killed by drugs in a monophasic manner [78,131,133]. To quantify the biphasic nature of shallow inhibition, they developed a biphasic model represented by the equation below.
I = F_1_ × [D]/([D] + K_d1_) + F_2_ × [D]/([D] + K_d2_)

In this model, the inhibition (I) by a drug has two phases: F_1_ and F_2_ as fractions of total cell viability (F_1_ + F_2_ = 100%), and each phase has its own binding affinity (K_d1_ and K_d2_). Curve-fitting shallow dose-response data to this equation yields F_1_, F_2_, K_d1_, and K_d2_. This analysis reveals the relative role a potential driver plays in the viability of a multi-driver cancer cell (F_1_) and the potency of the driver inhibition (K_d1_). Thus, this analysis allowed the identification of drugs for each driver in a multi-driver cancer. 

Different drug dose-response patterns by mono-driver and multi-cancer driver cancer cells are illustrated in Figure 2. In a mono-driver cancer cell (Figure 2A), a drug blocking the driver causes a mono-phasic dose-response pattern (Figure 2B). In a multi-driver cancer cell (Figure 2C), a drug blocking one driver would cause only a partial inhibition, such as curve 1 in Figure 2D. Sometimes, a drug may cause additional off-target inhibition, generating a biphasic curve as illustrated as curve 2 in Figure 2D. When inhibitors for different drivers are combined, each inhibitor is blocking its own target, and the combination would simultaneously block both drivers, leading to synergistic inhibition of cell viability, generating a monophasic like dose-response curve. This prediction has been confirmed in numerous colorectal cancer models [131,134] and TNBC cancer cells [78,115]. 

Applying this strategy to TNBC, highly potent drug combinations for the cell lines MDA-MB-231 and MDA-MB-468 were identified. Shen et al. determined that MDA-MB-231 contained two oncogenic drivers, Src and KRAS (due to a G13D mutation), and the cell proliferation can be partially inhibited by either the Src inhibitor dasatinib or Mek inhibitors trametinib or selumetinib. Each drug alone causes a shallow and biphasic inhibition of MDA-MB-231 cells and does not lethally inhibit MDA-MB-231; however, the combination of dasatinib and trametinib can lethally inhibit MDA-MB-231 cells with an IC_50_ of 8.2 nM. The combination also displays striking synergy. For example, the IC_70_ (drug concentration for 70% inhibition) is 25 nM for the combination, 12.6 μM for dasatinib, and above 20 μM for trametinib, resulting in a combination index (CI) of <0.003, and a dose reduction index (DRI) > 300 [135,136]. Thus, the combination is >300-fold more potent than dasatinib/trametinib without synergy. 

The same approach also identified a potent drug combination for MDA-MB-468. It was demonstrated that MDA-MB-468 proliferation and survival is dependent on EGFR over-expression and the activated PI3K pathway due to low expression of PTEN [78]. The cells are partially sensitive to both lapatinib (an EGFR inhibitor) [137,138] and GSK690693 (an Akt inhibitor) [139,140], and are potently inhibited by the lapatinib/GSK690693 combination (IC_50_ = 22 nM). The drug combination is also strikingly synergistic, with a CI of 0.025 and DRI of 40 at 70% inhibition [78]. 

The combinations are also strikingly specific for MDA-MB-231 and MDA-MB-468. Dasatinib + trametinib is 1800-fold more potent to MDA-MB-231 than MDA-MB-468 (IC_50_ of 8.2 nM vs. 15 μM), and lapatinib + GSK690693 is 454-fold more potent to MDA-MB-468 than MDA-MB-231 (IC_50_ of 22 nM vs. 10 μM). The specificity indicates that the inhibition is mechanism-based. This strategy of developing combination targeted therapy is yet to be verified in animal models and in clinical settings, but it offers a strategy to identify potent, synergistic, and mechanism-based targeted drug combinations for multi-driver cancers, such as TNBC.

## 6. New and Emerging Targets and Treatments for TNBC

New targets and new targeting strategies are emerging from preclinical studies of TNBC. These include targeting metabolism, epigenetic regulation, and developing new methods to target proteins and enzymes. These strategies could provide new treatment options in the future for TNBC patients. 

### 6.1. Targeting Metabolism

One emerging hallmark of cancer is reprogrammed energy metabolism [141], where cancer cells greatly increase glucose uptake and become more reliant on hyperactivated glycolysis and less reliant on decreased oxidative phosphorylation for energy production. As such, one area of potential targets are metabolic enzymes in glycolysis, oxidative phosphorylation, and lipid metabolism. The reprogrammed metabolism not only promotes cell proliferation but also contributes to drug resistance to chemotherapy. Thus, targeting metabolism has the potential both as a direct therapeutic approach and as a tool to counter drug resistance.

Multiple targets have been studied to manipulate glycolysis, such as inhibiting glucose transporters (e.g., small molecules WZB117 and resveratrol) or glycolysis (e.g., 2-deoxy-D-glucose [142] as a glucose analog, metformin inhibiting hexokinase, and 3-bromopyruvate inhibiting glyceraldehyde-3-phosphate dehydrogenase). Even though oxidative phosphorylation is reduced in cancer cells, it still plays an essential role in TNBC cells, which makes oxidative phosphorylation a potential therapeutic vulnerability [143]. IAC-10759 is a novel inhibitor of complex I of the mitochondrial electron transport chain [144]. It inhibits the growth of a broad range of patient-derived xenograft TNBC tumors [143], and it is being tested for its therapeutic efficacy against TNBC and other solid tumors in a clinical trial (NCT03291938).

Other metabolic functions may also provide therapeutic targets for TNBC. Recent evidence demonstrates that upregulation of essential lipogenic enzymes acetyl-CoA carboxylase-α and fatty acid synthase enhances the malignant behavior of TNBC [144,145,146,147]. Mitochondrial morphology and dynamics have also been associated with TNBC tumor growth and metastasis [145,148].

### 6.2. Epigenetic Therapy

Epigenetic regulation of gene activity is exerted by a number of mechanisms, such as DNA methylation, histone Lys acetylation and methylation, and several forms of non-coding RNAs [149]. With the development and progression of a tumor, there is a progressive loss of total DNA methylation, an increase of hypermethylated CpG islands, and an increased histone modification [150,151,152]. Hypermethylation of the CpG islands causes the inactivation of numerous tumor suppressor genes, including BRCA1 [153,154]. In TNBC, epigenetic modifications are known to play a crucial role in the epithelial-mesenchymal transition (EMT) and metastasis [155,156]. Numerous anti-cancer therapeutic targets in epigenetic modifications have emerged, and small molecule inhibitors for enzymes in maintaining DNA methylation and histone modifications are being actively studied as therapeutic approaches to counter oncogenic processes (reviewed in [157,158]). Epigenetic therapy could prove particularly attractive for TNBC because of the lack of alternative targeted therapies.

### 6.3. New Therapeutic Modalities

In addition to the traditional enzyme inhibitors and monoclonal antibodies that make up most current targeted therapeutics, other therapeutic modalities, such as small molecules disrupting protein-protein interaction and Proteolysis Targeting Chimeras (PROTACs) could provide new ways to manipulate oncogenic molecular processes. These new intervening modalities significantly expand the number of targets that can be manipulated chemically.

A large number of cellular processes are regulated or mediated by protein–protein interaction, and targeting protein–protein interaction has been recognized as a valuable approach to manipulate cancer cell biology [159]. For example, p53 is a tumor suppressor lost in most TNBC tumors due to mutation (84%) or other pathway inactivating events, such as gain of MDM2 (14%) [3]. Restoring p53 function could be a useful approach to inhibit cancer progression, but enzyme inhibitors or monoclonal antibodies are not useful for this purpose. Small molecules activating p53 or blocking its interactions with MDM2 are able to restore p53 pathway function and make cancers more sensitive to anti-mitotic drugs [160,161]. Of note is the 2-sulfonylpyrimidine compound PK11007, which alkylates p53 on specific residues and increases p53 thermal stability. PK11007 inhibits cell proliferation, induces apoptosis, and alters the expression of genes involved in cell death in TNBC cells [162].

Another approach for targeting the non-enzymatic functions of proteins in cancer cells is the usage of PROTACs [163,164,165]. PROTACs are heterobifunctional small molecules consisting of two ligands joined by a linker: one ligand binds a target protein while the other binds an E3 ubiquitin ligase. Simultaneous binding of the target protein and ligase by the PROTAC induces ubiquitylation of the target protein and its subsequent degradation by the ubiquitin–proteasome system [165]. This technology allows specific degradation of the target protein. A number of PROTACs are in clinical trials for cancer therapy [165].

## 7. Concluding Remarks

Considerable effort has gone into understanding the genetic and biochemical mechanisms of TNBC development and into developing targeted therapies for its treatment. While immunotherapy, PARP inhibitor therapy, and antibody-drug conjugate topoisomerase inhibitor therapy have made significant advances, only a small portion of TNBC patients meet the genetic requirements for these treatments. Kinase signaling-based targeted therapy is plagued by two main issues: heterogeneity and multi-driver tumorigenesis. Because TNBC is notoriously heterogeneous, any targeted therapy is likely to be effective for only a small portion of TNBC cases. This obstacle can be overcome by biomarker-guided patient selection, which requires a better understanding of the biomarker–drug response relationship. The multi-driver nature of TNBC dictates that monotherapy blocking any one driver will not be effective for most TNBC cases. Instead, drug combinations simultaneously blocking multiple drivers are required. Developing effective synergistic combination targeted therapy for TNBC also requires a better definition of oncogenic driving mechanisms in TNBC. Several new and promising therapeutic approaches are in different stages of research and development, such as targeting cancer energy metabolism, epigenetic therapy, targeting protein–protein interaction and stability, and targeted protein degradation. These new strategies could lead to novel therapeutics for TNBC in the future.

## Figures and Tables

**Figure 1 biomolecules-13-01207-f001:**
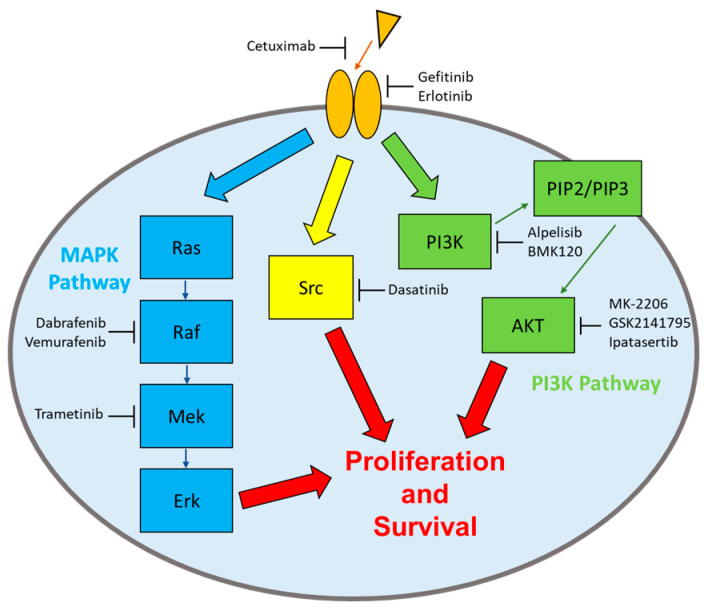
Signaling pathways and their respective targeted drugs. Four types of signaling enzymes have been the main targets of signaling drugs in TNBC: the rPTKs, Src, the MAPK pathway, and the PI3K pathway. The target enzymes and the drugs discussed are indicated.

**Figure 2 biomolecules-13-01207-f002:**
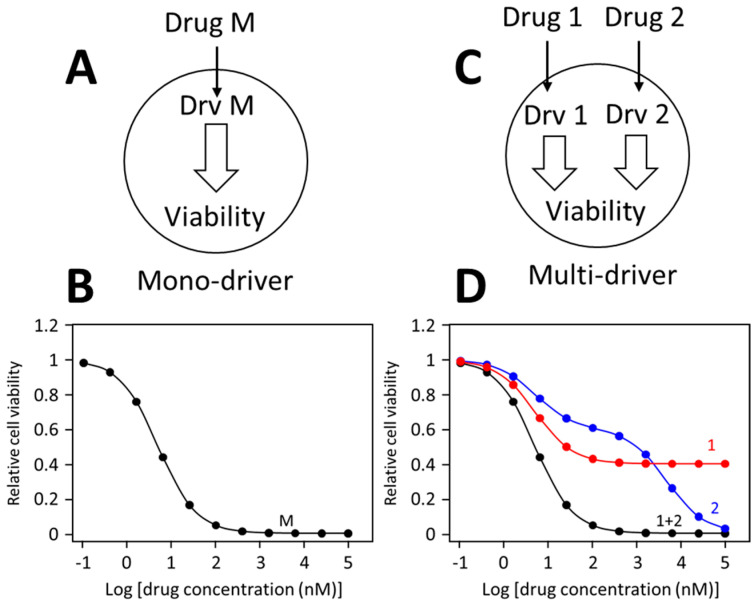
Dose-response curves of mono-driver and multi-driver cancer cells to targeted drugs. (**A**) Depiction of a mono-driver cell. (**B**) A monophasic dose-response curve by the mono-driver cancer cell to an inhibitor blocking its driver (Drv M). (**C**) Depiction of a multi-driver cancer cells with driver 1 (Drv 1) driver 2 (Drv 2). (**D**) Response of the multi-driver cancer cell to inhibitor for each driver individually or in combination. These dose-response curves illustrate idealized response patterns.

**Table 1 biomolecules-13-01207-t001:** FDA-approved targeted therapies for TNBC.

Drug	Target	Requirements
Oliparib	PARP	BRCA1 or BRCA2 mutation
Talazoparib	PARP	BRCA1 or BRCA2 mutation
Pembrolizumab	PD-1	Mismatch repair defect or microsatellite instability
Atezolizumab	PD-L1	Mismatch repair defect or microsatellite instability
Sacituzumab govitecan	Topoisomerase I	Two or more prior systemic therapies

**Table 2 biomolecules-13-01207-t002:** Clinical trials of signaling-based targeted therapy for TNBC.

NCT ID	Drug	Target	CR *	PR *	SD *	DP *	References
01629615	BMK120	PI3K	0/50 (0%)	0/50 (0%)	17/50 (34%)	20/50 (40%)	-
02506556	Alpelisib	PI3K	0/10 (0%)	0/10 (0%)	5/10 (50%)	2/10 (20%)	[99]
01277757	MK-2206	Akt	0/9 (0%)	0/9 (0%)	1/9 (11%)	8/9 (89%)	[100]
00371254	Dasatinib	Src	0/43 (0%)	2/43 (5%)	12/43 (28%)	22/43 (51%)	[107]
00817531	Dasatinib	Src	0/22 (0%)	2/22 (9%)	15/22 (68%)	5/22 (23%)	-
02720185	Dasatinib	Src	1/5 (20%)	NR *	NR *	NR *	-

* The clinical response was assessed using RECIST and based on the changes in the longest diameter of the target lesion measured. Complete Response (CR), disappearance of the target lesion; Partial Response (PR), ≥30% decrease in the diameter of target lesion compared to baseline; Progressive disease (PD), ≥20% increase in the diameter of target lesion, taking as reference the smallest diameter recorded since the baseline measurement or the appearance of new lesion; Stable disease (SD), neither sufficient shrinkage as PR nor sufficient increase as PD. NR: not reported.

## Data Availability

Not applicable.

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
