# Peer review of "Challenges and Opportunities in Developing Targeted Therapies for Triple Negative Breast Cancer"

_biomolecules, 2023, doi:10.3390/biom13081207_

Round 1
Reviewer 1 Report
Chapdelaine and Sun presented a review article that has focused on triple-negative breast cancer (TNBC) as a heterogeneous disease with likely multiple oncogenic drivers. They discussed a combination approach either with two targeted drugs or a targeted therapy and chemotherapy for TNBC treatment.
The followings need to be addressed:
1. Recent progress on TNBC therapy with the antibody-drug conjugate sacituzumab govitecan-hziy should be discussed.
2. Check the reference content or accuracy such as #21, 24.
3. Page 1 line 25, remove “Metastatic”.
Author Response
Reviewer 1:
- Recent progress on TNBC therapy with the antibody-drug conjugate sacituzumab govitecan-hziyshould be discussed.
Response: In response to this excellent suggestion, we added the section 2.3. Antibody-drug conjugate sacituzum govitecan.
- Check the reference content or accuracy such as #21, 24.
Response: We deleted the original reference 21. Reference 24 (#23 in the revised manuscript is spelled out as American Cancer Society in the revised manuscript.
- Page 1 line 25, remove “Metastatic”.
Response: The word is deleted as requested.
Thank you.
Reviewer 2 Report
This review is focused on developing strategies for the treatment of TNBC. It provides a comprehensive overview of existing and developing agents and analyzes data from the current clinical trials. The review provides a holistic view and highlights challenges in the field. Referencing is appropriate, and the material is well organized. The number of tables is appropriate. However, a few edits are warranted to increase the quality of the paper.
1. New targets for TNBC include pathways that target mitochondrial pathways (for example, OXPHOS) and glycolytic and lipogenic metabolism. The authors should include agents that target these pathways and acknowledge the potential of these pathways in developing TNBC.
2. A summary figure/scheme with the mentioned targets/pathways will make it easier for readers to work with the review.
Author Response
Reviewer 2:
- New targets for TNBC include pathways that target mitochondrial pathways (for example, OXPHOS) and glycolytic and lipogenic metabolism. The authors should include agents that target these pathways and acknowledge the potential of these pathways in developing TNBC.
Response: In response to this excellent suggestion, we added a section discussing approaches of targeting glycolysis, oxidative phosphorylation and lipogenic enzymes. See section 6.1. Targeting metabolism.
- A summary figure/scheme with the mentioned targets/pathways will make it easier for readers to work with the review.
Response: We added a Figure 1 that includes all the signaling targets and drug candidates discussed in this review. The original Figure 1 is changed into Figure 2.
Thank you.
Reviewer 3 Report
This is an ambitious article that attempts to summarize the current status of research on targeted therapeutic agents for triple-negative breast cancer and the current opportunities and challenges. This is a good topic that will attract a lot of experts in the field of breast cancer research. However, this article summarizes some targeted drugs currently in clinical testing, and does not summarize some newly discovered targets, such as epigenetic drugs, protein-protein interaction drugs, targeted degradation drugs, and so on (BRD4, KDM5A, Wee1...). . This will greatly reduce the comprehensiveness of the article and reduce the audience. I therefore suggest that the author add this section to the main text and discussion section. In addition, some following minor revisions are also needed.
1. The table in the text should use a standard three-line table.
2. Abbreviations need to be defined when they first appear.
3. A table is needed to help understand the section "Approved targeted therapy options for TNBC patients".
4. The names of nucleotide sequences such as gene names should be italicized when appearing in articles and transcripts.
5. The first letter of each word should be in uniform for Refferences.
Overall , I recommend major revision for this manuscript.
Minor editing of English language required
Author Response
Reviewer 3:
This is an ambitious article that attempts to summarize the current status of research on targeted therapeutic agents for triple-negative breast cancer and the current opportunities and challenges. This is a good topic that will attract a lot of experts in the field of breast cancer research. However, this article summarizes some targeted drugs currently in clinical testing, and does not summarize some newly discovered targets, such as epigenetic drugs, protein-protein interaction drugs, targeted degradation drugs, and so on (BRD4, KDM5A, Wee1...). . This will greatly reduce the comprehensiveness of the article and reduce the audience. I therefore suggest that the author add this section to the main text and discussion section. In addition, some following minor revisions are also needed.
Response: In response to this excellent suggestion, we added a section discussing several newly discovered targets and therapeutic approaches, including targeting metabolism (section 6.1), epigenetic therapy (section 6.2), targeting protein-protein interaction and stability (section 6.3) and targeted protein degradation (section 6.3). These emerging approaches are mostly in pre-clinical development or early stages of clinical development. They have not been proven effective yet but hold considerable promise for the future. There is considerable amount of work in each of these areas, but the scope of this current review limits us to a very superficial survey of these topics to highlight their general mechanism and future promise. We hope that this treatment meets the approval of the reviewer.
- The table in the text should use a standard three-line table.
Response: We are not sure how to specifically revise the table. In our understanding, the format of the table in this article is a three-line table. We are willing to let the journal to format it in any way to fit the style of the journal.
- Abbreviations need to be defined when they first appear.
Response: We went through the manuscript to add abbreviations when they first appeared.
- A table is needed to help understand the section "Approved targeted therapy options for TNBC patients".
Response: We added a table 1, summarizing the approved therapies. The original Table 1 is now renumbered as Table 2.
- The names of nucleotide sequences such as gene names should be italicized when appearing in articles and transcripts.
Response: We italicized gene names.
- The first letter of each word should be in uniform for Refferences.
Response: I hope that the journal will take care of formatting issues like this.
Overall, I recommend major revision for this manuscript.
Response: Thank you for the excellent suggestions which significantly improved the manuscript.
Reviewer 4 Report
In this review article, Chapdelaine et al. discuss the current treatment options for triple negative breast cancer (TNBC), which primarily include surgery, radiation therapy, and chemotherapy. The development of targeted therapies for TNBC is a challenge due to the heterogeneity of the disease. The article highlights that TNBCs are considered multi-driver cancers, where multiple oncogenic drivers contribute to cell proliferation and survival. Therefore, effective targeted therapies for TNBC would involve combinations of drugs that simultaneously target multiple oncogenic drivers. The article also presents a strategy for generating mechanism-based combination therapies. Overall, the text is well-written and provides a clear overview of the current therapeutic approach and potential future directions for TNBC patients. To enhance the reader's understanding, it might be helpful to include a graphical scheme illustrating inhibitors and their molecular targets, allowing for better visualization of therapeutic strategies.
Author Response
Reviewer 4:
In this review article, Chapdelaine et al. discuss the current treatment options for triple negative breast cancer (TNBC), which primarily include surgery, radiation therapy, and chemotherapy. The development of targeted therapies for TNBC is a challenge due to the heterogeneity of the disease. The article highlights that TNBCs are considered multi-driver cancers, where multiple oncogenic drivers contribute to cell proliferation and survival. Therefore, effective targeted therapies for TNBC would involve combinations of drugs that simultaneously target multiple oncogenic drivers. The article also presents a strategy for generating mechanism-based combination therapies. Overall, the text is well-written and provides a clear overview of the current therapeutic approach and potential future directions for TNBC patients. To enhance the reader's understanding, it might be helpful to include a graphical scheme illustrating inhibitors and their molecular targets, allowing for better visualization of therapeutic strategies.
Response: We appreciate the positive comments from the reviewer. In response to the suggestion and a similar suggestion from another reviewer, we added a new Figure 1 that illustrates the main oncogenic pathways in TNBC the drug candidates targeting them. The original Figure 1 is now Figure 2.
Thank you.
Round 2
Reviewer 3 Report
The authors have addressed all my concerns. I recommend recepting this manucript.